# A Matter of Approach: Analysis of the Flow Refuge Preferences of Iberian Barbels during Pulsed Flows in Flume Conditions †

Renan Leite [1,2,*], Maria João Costa [1,*], Anthony Merianne [1,*,‡], Daniel Mameri [2,*], José Maria Santos [2,*], Antonio Nascimento Pinheiro [1,*] and Isabel Boavida [1,*]

1. Civil Engineering for Research and Innovation for Sustainability, Instituto Superior Técnico, Campus of Alameda, University of Lisbon, 1049-003 Lisbon, Portugal
2. Forest Research Centre (CEF) and Associate Laboratory TERRA, School of Agriculture, University of Lisbon, Tapada da Ajuda, 1349-017 Lisbon, Portugal
* Correspondence: renanleite@edu.ulisboa.pt (R.L.); mariajcosta@tecnico.ulisboa.pt (M.J.C.); anthony.merianne@grenoble-inp.org (A.M.); dmameri@isa.ulisboa.pt (D.M.); jmsantos@isa.ulisboa.pt (J.M.S.); antonio.pinheiro@tecnico.ulisboa.pt (A.N.P.); isabelboavida@tecnico.ulisboa.pt (I.B.)
† Presented at the IX Iberian Congress of Ichthyology, Porto, 20–23 June 2022.
‡ Presenting author (oral communication).

**Abstract:** To fight against global warming, we have to change our ways of consuming energy. Due to its low carbon impact and strong dispatchability, hydroelectric production will be one of the bases of this transition. However, peak electricity demand produces rapid and artificial flow fluctuations in tailwaters, i.e., hydropeaking, which has negative effects on fish biota. Thus, developing effective mitigation measures against hydropeaking is an urgent matter. The present study aims to limit the impact of this flow fluctuation on an Iberian cyprinid fish: the Iberian barbel (*Luciobarbus bocagei*). We experimentally tested different angles of flow refuge entrances (45° and 70°) in an indoor flume (6.5 m × 0.7 m × 0.8 m) to determine if this would affect the behavior of the fish. For each angle configuration, two refuges were installed and distanced 2.30 m from each other on the same side of the flume. Three possible resting locations were defined: downstream, inside, or upstream of each structure. Both angles were tested at 7 L/s (base flow), simulating the normal conditions of the river, and 60 L/s (peak flow), simulating a hydropeaking event. Each replicate comprised a group of five fish. For each, the frequency and residence time were quantified. The preliminary results indicated that the fish movement patterns changed when peak flow occurred. The downstream refuge was more frequently used in both configurations during peak flow. Additionally, the inside parts of the refuges were more frequently used, instead of the parts immediately downstream or upstream, and the time spent inside the refuge at peak flow was higher when compared to base flow. Additionally, hydraulic experiments were carried out at each configuration to determine the velocity field using ADV (Vectrino) technology. For the base flow, mean water depth and average velocity were 8 cm and 12 cm/s, respectively, increasing to 24 cm and 39 cm/s during peak flow. Measurements showed that velocity was equal to 74 cm/s in the narrowed area due to the refuge location, and velocity was null inside and directly downstream of the refuge. The results from this study will allow the development of guidelines for designing flow refuges for cyprinid fish, and hence mitigate the impact of hydropeaking.

**Keywords:** hydropeaking; hydropower plants; cyprinids; flow refuges; velocity

**Author Contributions:** Conceptualization, I.B. and M.J.C.; Methodology, I.B. and M.J.C.; Software, A.M. and R.L.; Validation, I.B., M.J.C., J.M.S. and A.N.P.; Formal analysis, I.B., R.L., M.J.C. and A.M.; Investigation, I.B., M.J.C., R.L., D.M. and A.M.; Resources, I.B., M.J.C., J.M.S. and A.N.P.; Data curation, I.B. and M.J.C.; Writing-original draft preparation, R.L. and A.M.; Writing-review and editing, I.B., M.J.C., J.M.S., R.L., D.M. and A.M.; Visualization, I.B. and M.J.C.; Supervision, I.B. and M.J.C.; Project Administration, I.B. and M.J.C.; Funding acquisition, I.B. All authors have read and agreed to the published version of the manuscript.

**Funding:** This research was funded by FCT "Fundação para a Ciencia e a Tecnologia", grant number: PTDC/EAM-AMB/4531/2020. The authors are grateful for the Foundation for Science and Technology's support through funding UIDB/04625/2020 from the research unit CERIS.

**Institutional Review Board Statement:** The Institute for Nature Conservation and Forests (ICNF) provided the necessary fishing and handling permits, and General Directorate of Food and Veterinary Medicine (DGAV) provided the authorization to perform experimental research with live animal. All research was conducted in accordance with national and international guidelines for animal welfare.

**Informed Consent Statement:** Not applicable.

**Data Availability Statement:** Data is available upon request from the authors.

**Conflicts of Interest:** The authors declare no conflict of interest.