# Peer review of "A Matter of Approach: Analysis of the Flow Refuge Preferences of Iberian Barbels during Pulsed Flows in Flume Conditions†"

_blsf, doi:10.3390/blsf13010086_

Round 1

Reviewer 1 Report

  • Lines 22-25: The sentence should be rewritten for better understanding.
  • Line 32-33: It's difficult to understand this sentence without seeing the figure.
  • Line 37: 24 cm/s.

Author Response

Lines 22-25 : changed to 

"By experimentally testing in an indoor flume (6.5m x 0.7m x 0.8m) different angles of flow refuge entrance – 45º and 70º - experiments aims to determine if it would affect their use by fish. For each angle configuration, two refuges were installed and distanced 2.30m from each other on the same side of the flume. Three possible resting locations were defined: downstream, inside or upstream of each structure"

Lines 32-33 : changed to 

"Also, inside part of refuges were more frequently used instead of immediately downstream or upstream of it, and the time spent inside the refuge at peak flow was higher when compared to base flow. "

Line 37 : The 24cm corresponds to the water depth for peak flow discharge, so the units should still be cm.